# PH Responsive Polyurethane for the Advancement of Biomedical and Drug Delivery

**DOI:** 10.3390/polym14091672

**Published:** 2022-04-20

**Authors:** Rachel Yie Hang Tan, Choy Sin Lee, Mallikarjuna Rao Pichika, Sit Foon Cheng, Ki Yan Lam

**Affiliations:** 1School of Postgraduate, International Medical University, Kuala Lumpur 57000, Malaysia; rachel.tanyie@student.imu.edu.my (R.Y.H.T.); lam.kiyan@student.imu.edu.my (K.Y.L.); 2Department of Pharmaceutical Chemistry, School of Pharmacy, International Medical University, Kuala Lumpur 57000, Malaysia; mallikarjunarao_pichika@imu.edu.my; 3Centre for Bioactive Molecules and Drug Delivery, Institute for Research, Development and Innovation, International Medical University, Kuala Lumpur 57000, Malaysia; 4Unit of Research on Lipids (URL), Department of Chemistry, Faculty of Science, University of Malaya, Kuala Lumpur 50603, Malaysia; sfcheng@um.edu.my

**Keywords:** pH-responsive, polyurethane, biomedical, drug delivery

## Abstract

Due to the specific physiological pH throughout the human body, pH-responsive polymers have been considered for aiding drug delivery systems. Depending on the surrounding pH conditions, the polymers can undergo swelling or contraction behaviors, and a degradation mechanism can release incorporated substances. Additionally, polyurethane, a highly versatile polymer, has been reported for its biocompatibility properties, in which it demonstrates good biological response and sustainability in biomedical applications. In this review, we focus on summarizing the applications of pH-responsive polyurethane in the biomedical and drug delivery fields in recent years. In recent studies, there have been great developments in pH-responsive polyurethanes used as controlled drug delivery systems for oral administration, intravaginal administration, and targeted drug delivery systems for chemotherapy treatment. Other applications such as surface biomaterials, sensors, and optical imaging probes are also discussed in this review.

## 1. Introduction

The applications of stimulus-responsive polymers in the biomedical and drug delivery fields have been gaining much attention in recent years. Stimulus-responsive polymers are known as smart, intelligent or stimuli-sensitive polymers. They can undergo reversible physical or chemical changes in response to external changes in environmental conditions, and the changes can be observed in properties including surface activity, chain conformation, solubility, configuration and morphology [1]. The examples of different classes of stimulus responsiveness are shown in Figure 1, which include biochemical responsiveness such as enzyme- and glucose-responsive, chemical responsiveness such as pH-, redox-, and solvent-responsive and physical responsiveness such as temperature-, light-, and mechanical-force-responsive.

Stimulus-responsive polymers are categorized into single-responsive, dual-responsive and multi-responsive properties. Single-responsive polymers can only be responsive to one stimulus, whereas dual- or multi-responsive polymers can respond to several stimuli. With different responsiveness, the polymers can also act differently as a result of changing environment conditions. These polymers have been studied over the years for a range of medical, biomedical and drug delivery applications, such as oral administration drug delivery systems [2], intravaginal drug delivery systems [3], anticancer drug delivery systems [1,4], tissue engineering [5], biomaterials such as sensors and actuators [1,6], gene delivery systems [7] and smart coatings [8,9].

### pH-Responsive Polymers

pH-responsive polymers are polymers that respond to changes in environmental pH. They can be classified into: (A) polymers with ionizable moieties; and (B) polymers that contain acid-labile linkages.

Polymers containing ionizable functional groups are prepared by introducing pendant acidic groups or basic functional groups during the polymerization reaction [1,10,11]. The ionizable moieties either protonate or deprotonate at different pH values. For high pH-responsive polymers, acidic functional groups (polyanions) such as carboxylic acid, sulfonic acid or methacrylic acid groups of the polymers are conjugated in the polymer backbone (Figure 2A). These polymers will dissociate and deprotonate at high pH conditions, resulting in higher charge density due to the electrostatic repulsion between the chains that further leads to absorption of water, hence the polymer swells and the incorporated content is released (Figure 2B). On the other hand, the low pH-responsive polymers required basic functional groups (polycations) such as pyridine derivatives, piperazines or amino salt groups (Figure 3A). The basic functional groups are ionized and protonated at low pH conditions, increasing the charge repulsion between neighboring polybasic groups and expanding the network; hence the polymer swells and the incorporated content is released (Figure 3B).

Amphoteric polymers, also known as zwitterionic polymers, are polymers having both acidic pendant groups (anionic) and basic pendant groups (cationic), which are capable of swelling in both acidic and basic media (Figure 4A) [12,13]. The electrostatic interaction can be varied by changing the environmental pH or modifying the ion concentrations of the interfaces. Luo et al., (2010) reported on the synthesis of novel amphoteric pH-sensitive hydrogels derived from ethylenediaminetetraacetic dianhydride, butanediamine and amino-terminated poly(ethylene glycol) [14]. In the study, both –NH^+^ cations and –COO^−^ anions were at equilibrium at pH 7, which induced electrostatic attractions and ionic cross-linking to the polymer, hence limiting the swelling rate. When the medium was altered to pH 2 or pH 11, there were only –NH^+^ cations and –COO^−^ anions, respectively. The strong electrostatic repulsion induced water diffusion into the network, leading to anomalous diffusion. The mechanism of action was similarly reported by Manal et al., in which a hydrogel containing both methacrylic acid and N,N-dimethyl amino ethyl methacrylate (DMAEMA) was polymerized for insulin delivery [15]. The hydrogels swelled at pH < 3.0 and pH > 6.0. Between pH 3.0–5.0 (isoelectric point), electrostatic attraction happens between NH_3_^+^ cations and COO^−^ anions (which dominates over repulsion between unpaired like charges), leading to decreased swelling (Figure 4B).

Polymers containing acid-labile linkages (Figure 5) are prepared by introducing linkers such as hydrazone, acetal, imine and ortho ester into the backbone of the polymer [16]. By decreasing the environmental pH, the hydrolysis process is triggered and cleaves these bonds, causing degradation of the polymer chain (Figure 6); the drug is then released. Polymers with acid-labile linkages have been used in developing anticancer drug delivery systems. These polymers are relatively stable in neutral and basic media, but labile in an acid medium due to their covalent linkage, which has slower internal structural transition compared to polymers with ionizable moieties.

pH-responsive polymers are useful in biological applications due to the variation of pH throughout the body. For instance, the stomach has an acidic condition of around pH 2, while the small intestine has a relatively basic condition, around pH 7.2 to 7.5. In addition, cancer cells have a lower pH value (pH 6.0 to 6.5) compared with normal tissues (pH 7.0) [17,18]. This is due to excess accumulation of lactic acid produced by the glycolysis rate in the pathological tissues. Owing to these remarkable variations of pH, along with different tissue sites and sub-cellular spaces, pH-responsive polymers have been used to treat these physiological features in medical and biomedical applications. The mechanism of pH-responsive polymers in drug release systems depends on their swelling and contraction behavior or their degradation properties (cleavage of functional groups). Hence, they have been studied for their applications in oral drug delivery systems of sulforhodamine B [19], gene delivery systems [20] and as biosensors for medical diagnostics or health monitoring [21].

Table 1 categorizes studies that reported on pH-responsive polymers by their mechanism: (A) ionizable moieties; and (B) acid-labile linkages and their applications, for the past two years. Polymers with ionizable functional groups, most of which contained pendant basic groups, especially from amines and their derivatives, are shown in Figure 7. These polymers respond to acidic pH ranging from pH 4.0–6.5; the polymers swell, resulting in enhanced drug release. This group of polymers was used in chemotherapy drug delivery, gene delivery, and certain local therapies. These polymers could respond to an acidic environment, which causes swelling and results in rapid drug release at specific tumor sites. There are also polymers containing pendant carboxylic acid groups (Figure 8), in which the polymers swell and enhance release at pH 7.4.

On the other hand, polymers containing acid-labile linkage, such as β-thiopropionate, hydrazone linkage, oxazaline and boron-ester linkage, all respond to acidic pH ranging from pH 5.0–6.8, resulting in increased drug release. The applications for this class of polymers were mostly for delivery of anti-cancer drugs, which have non-specific distribution and uncontrollable release. From the results of studying the response to pH value of acid-labile linkages, these polymers aid in the specific target release of anticancer drugs at the tumor cells (pH ranging from 6.0–6.5). Acid-labile linkages are used for their tunable properties, as they can be incorporated into block copolymers at various positions, including the hydrophobic block backbone, pendant chains or at the hydrophilic/hydrophobic block junctions. They remain stable at blood pH, while they degrade upon cleavage under acidic pH conditions (tumor site).

Many patents and academic papers have been published on potential applications of pH-responsive polymers in drug delivery systems and biomedical devices, however only a few have successfully been commercialized. One of the known commercial pH-dependent polymers is the Eudragit series of methacrylate copolymers, including Eudragit E (applied in sublingual and topical preparations for immediate release at pH < 5), Eudragit S100, Eudragit L100, Eudragit L100-55 (applied in enteric coating for delayed release at pH > 5.5) and Eudragit SF 30 D (applied in colonic drug delivery systems) [22]. Other commercially available pH-dependent polymers are cellulose acetate phthalate (CAP), polyvinyl acetate phthalate (PVAP), hydroxyl propyl methyl cellulose (HPMCP), cellulose acetate trimelliate and Kollicoat MAE 30 DP, in which these polymers are commonly used as enteric coating for solid dosage form [23]. SQZ Gel^TM^ is a marketed pH-sensitive chitosan-and-polyethylene-glycol-blend hydrogel for controlled release formulations. Most of the developed systems remained at the preclinical stage, due to the various conditions of patients which could interfere with the accuracy of results in the clinical stage. Thus, before being commercialized, further investigations and trials ought to be carried out.

**Table 1 polymers-14-01672-t001:** pH-responsive polymers categorized by the mechanism (A) Ionizable moieties and (B) Acid-labile linkages, and their application.

Mechanism	Pendant Groups	Ionizable Group/Linkage	Responsive pH	Polymer Type	Type of Response	Application	Ref
**(A)** **Ionizable Moieties**	**Basic pendant groups, such as amine and its derivatives**	2-aminoethyl methacrylate	6.5 and 6.8	Polyplex Nanoparticles (NPs)	Improve cellular uptake and transfection efficiency	Gene Delivery	[7]
		2-methoxy-4-aminomethyl-1,3-dioxolan	5.5	Poly(vinyl alcohol) PVA	Rapid drug release	Drug carrier for tumor therapy	[24]
		Histidine	5.0	Poly(ethylene glycol)-polycaprolactone PEG-PCL	Increase drug release	Chemotherapy	[25]
		2-dimethylamino ethyl methacrylate	4.0 and 7.0	Poly(lactic acid) PLA	Swells and speeds drug release	Targeted drug delivery vehicles	[26]
		2-dimethylamino ethyl methacrylate	5.0	PCL	Swells and enhances drug release	Drug delivery	[27]
		Chitosan	5.5	Poly(lactic-co-glycolic acid) PLGA	Efficient release	Tacrolimus delivery	[28]
		Chitosan	5.5	Poly(N-isopropylacrylamide)-co-itaconic acid NIPAM	Fast release	Local breast cancer therapy	[29]
		2-(diisopropylamino) ethyl methacrylate (DPA)	6.5	Hydroxyethyl methacrylate-co-DPA copolymers	Increased drug release	Ocular drug delivery	[30]
	**Carboxylic acid pendant groups**	Methacrylic acid (MAA)	7.4	Amine-modified bimodal mesopores silica	Swells and speeds drug release	Drug delivery carrier	[31]
		Acrylic acid	7.4	Poly(ethylene glycol) PEG	Increased drug release	Targeted drug delivery	[32]
**Mechanism**	**Acid-Labile Groups**		**Responsive pH**	**Polymer Type**	**Type of Response**	**Application**	**Ref**
**(B)** **Acid-labile linkages**	β-Thiopropionate		5.0	PEG	Increased drug release	Targeted cancer cell treatment	[33]
	β-Thiopropionate		5.0	Poly(beta-thioether ester)-PEG	Rapid drug release	Nanocarrier for drug delivery	[34]
	Hydrazone linkage		5.5	Lipid Polymer Hybrid NPs	Fast drug release	Biomedical and chemotherapy	[17,35]
	Hydrazone linkage		5.4	N-isopropylacrylamide-co-glycidyl methacrylate (NIPAM-co-GMA)	Increased drug release	Chemotherapy drug delivery	[36]
	Hydrazone linkage		5.6	Poly(β-benzyl malate)	Rapid drug release	Antitumor drug carrier	[37]
	Oxazoline		6.0	Poly(lactic acid)-poly(β-amino ester)	Increased drug release	Colon cancer adjuvant therapy	[38]
	Boron-ester linkage		6.8	Polymer dots	Better fluorescent intensity	Bioimaging probe	[39]
	Borate-ester linkage		5.5	PNIPAAm Poly(N-isopropylacrylamide)	Rapid drug release	Cancer therapy	[40]
**Others**	Metal ligand (Fe^3+^)		5.0	PEG-PLGA	Rapid drug release	MRI-guided therapy	[41]

## 2. Stimulus-Responsive PU

### 2.1. Polyurethane

Polyurethane (PU) is a highly utilized synthetic polymer, used to manufacture rigid or flexible foams, coatings, adhesives, sealants, and elastomers. The unique feature of PU is the phase segmentation of hard and soft segments within its structure (Figure 9), which makes it very versatile. Moreover, it can be synthesized and conveniently modified into different forms of polymeric structures, ranging from elastomeric to porous, by altering the ratio of hard to soft segments. The soft segment is comprised of polyhydroxyl components, while the hard segment is composed of polyisocyanates. PU offers good abrasion resistance and toughness, thus driving its commercial applications in many industries. It is biodegradable, biocompatible and hemocompatible, and demonstrates an accommodative biological response when used in biomedical applications such as bone or tissue implants and controlled drug-release systems [12,42,43].

The conventional synthesis route of PU is through reacting polyhydroxyl compounds (–OH group) from non-renewable petrochemicals and polyisocyanates (–NCO group) (Figure 1). Polyhydroxyl compounds which have been widely used are polycaprolactone (PCL), polyethylene glycol (PEG), polytetramethylene ether glycol and hexamethylene glycol, which are mostly derived from non-renewable petrochemicals [44,45]. Due to the depletion of petroleum, biodegradable and renewable sources were introduced to replace the petrochemicals. Hence, many organizations and researchers are working towards producing bio-based polyurethanes developed from vegetable oils such as soybean oil [46], castor oil [47], sunflower oil [48], and linseed oil [49], for environmentally friendly and biodegradable considerations.

The most commonly used aromatic diisocyanates are toluene diisocyanate (TDI) and methylene bis-diphenyl isocyanate (MDI), which give better crosslinking; while the commonly used aliphatic polyisocyanates are hexamethylene diisocyanate (HDI) and hydrogenated MDI, which normally act as chain extenders [50,51]. Aromatic isocyanates are more reactive, which results in faster reactions. This is due to the presence of an electron-withdrawing group on the isocyanate molecule, which increases the electron density charge, resulting in easier electron transfer from the donor to the carbon [50]. However, aromatic structures are not desired for medical and biomedical applications as they can produce toxic degradation products [45]. In addition, polyisocyanates are classified as a carcinogenic substance by the Occupational Safety and Health Administration (OSHA), and many organizations are working hard to ban isocyanate-synthesized products. Furthermore, during the process, diamine phosgenation is performed in solvents such as chlorobenzene to produce diisocyanates. The solvents used and the phosgene gas produced during this process cause environmental concerns, and create an exposure risk hazard that could affect workers’ health.

Hence, in recent years, PU synthesis is being classified into isocyanate-based and non-isocyanate-based; while the polyols used can be also differentiated by petrochemical-based and vegetable oil-based. The non-isocyanate PUs were produced from different routes, which include: polyaddition of bifunctional cyclic carbonates with diamines [52]; polycondensation of ethylene carbonate, diamines and diols [52,53]; cationic ring-opening polymerization of cyclic urethanes [33,54]; and copolymerization of substituted aziridines with CO_2_ [55] (summarized in Figure 2).

### 2.2. Types of Stimuli/Stimulus-Responsive Polyurethanes Used in Biomedical and Drug Delivery Applications

Polyurethane has been introduced into stimulus-responsive materials; Table 2 summarizes different types of stimulus-responsive PUs that could potentially be applied in the biomedical and drug delivery system in the recent 3 years (database from Web of Science, June 2021). The PU types can be categorized into four types: (A) isocyanate-based PUs and (B) non-isocyanate-based PUs; (C) PUs synthesized with petrochemical-based polyols; and (D) PUs synthesized from bio-based polyols. Table 2 has excluded pH-responsive PUs since they are discussed in the subsequent section. PUs with single-, dual- and multi-responsiveness have been reported. Most of the PUs in the reported works are produced by reacting isocyanates and petrochemical-based polyols (Type A + C), whereby some PUs were purchased commercially, and only two works reported on ring-opening polymerization by reaction of carbonates and tyrosine with petrochemical polyols (Type B + C). There were no studies reporting on non-isocyanate-based responsive PU.

The single-stimulus-responsive PUs reported include shape memory PUs, thermo-responsive PUs, redox-responsive PUs and light-responsive PUs (Table 2). Thermo-responsive PUs were studied in drug delivery systems due to the variation in temperature between normal cells and inflamed cells. Shape memory PUs have better mechanical properties, force resistance and shape recovery properties, which are suitable for biomaterials applications such as sensors, actuators, vascular stents, and endovascular embolization. Redox-responsive PUs are more likely to be applied in drug delivery. This is due to the fact that within the body, there are different compositions of proteins and enzymes at specific sites which can create reduction or oxidation environments, which further lead to dissociation and drug release. For instance, glutathione (GSH) concentration is higher in tumor cells. By incorporating drugs into a redox-responsive polymeric system with disulfide bonds, the system can easily break down and reduce GSH into sulfhydryl groups, which causes self-degradation and release of the drug [56]. As for photo-responsive PUs, they are mostly designed for application as biosensors or drug carriers for cancer therapies.

The dual-stimuli-responsive PUs mainly consist of the combination of two stimuli to optimize the formulation and increase the effectiveness of its biomedical applications. Most of the studies reported for dual stimulus response come with a combination of thermo- and photo-responsive, thermo- and shape-memory-responsive, thermo- and enzyme- responsive or shape-memory-and water-responsive PUs (Table 2).

Biocompatibility and biodegradation evaluations are important for biomaterials used in medicine and drug delivery. A biodegradable polymer degrades into normal metabolites of the body or is eliminated from the body with or without further metabolic transformation. Biodegradation tests were conducted to monitor the breakdown status of the polymers and the breakdown rate of materials mediated by biological activity [57]. For instance, when exposed to bodily fluids, the materials may undergo chemical, physical, mechanical or biological changes, which may lead to different results in the body. By knowing the rate of degradation, the duration of the polymers persisting inside the body could also be estimated. The biodegradation test is important to achieving desired degradability, along with suitable sustainability [58]. The common biodegradation test was conducted by immersing polymers into a phosphate buffer solution (PBS) containing enzymes (lipase) at a certain pH value for hours to days.

The biocompatibility test confirms the suitability of a polymer when exposed to bodies or bodily fluids. A biocompatible polymer would not trigger any allergies or have side effects when used within the body [57]. The ideal biological research methodology consists of both in vivo and in vitro studies. For in vivo tests, the materials must be exposed to hemocompatibility, cytotoxicity, mutagenicity and pyrogenicity tests. However, due to ethical aspects, in vitro cytotoxicity tests were widely used to determine whether the material contains significant quantities of biologically harmful components. CCK8 assays, MTT assays and MTS assays were used in the reported studies for in vitro cytotoxicity tests.

Most of the studies included in Table 2 have conducted in vitro cytotoxicity and biodegradation studies to evaluate the PUs’ biocompatibility and biodegradability, except for the shape-memory-responsive PUs. For shape memory PUs, most of the studies conducted were mainly focusing on the polymers’ shape memory behavior and their tensile and mechanical ability to withstand force. For the biocompatibility evaluation, most of the reported studies showed high cell viability of more than 65%, which can be concluded as potentially biocompatible polymeric materials and safe to be used as biomaterials [59,60]. Other than interactions between the polymers and cells, the possible effect caused by toxicity of the degradation products must be evaluated. Although some of the studies claimed that PU degraded into non-toxic products or metabolites [61], the biocompatibility after the release of incorporated substances has to be taken into consideration. Hence, while conducting the biocompatibility studies, the biological and chemical properties as well as cytotoxicity of the degraded species should be investigated [62]. Furthermore, histological studies of in vivo implant sites could be conducted to determine the cytotoxicity of degraded stimulus-responsive PUs toward the surrounding tissues [63]. For biodegradability evaluation, weight loss of the PU polymers could be observed within days to weeks. pH-responsive PUs will be further discussed in the following section.

**Table 2 polymers-14-01672-t002:** Types of stimulus-responsive polyurethanes used in biomedical and drug delivery applications, and their biocompatibility and biodegradability evaluation (excluding pH-responsive).

Type of Stimuli/Stimulus-Responsive PU	Type of Stimuli/Stimulus	* PU Type	PU Name	Reactants for Synthesis of PU	Biocompatibility Evaluation	Biodegradability Evaluation	Applications	Ref
Single	Thermo	A + C	Poly(ether urethane) (PEU)	i. Poly(ethylene oxide)-poly(propylene oxide)-poly(ethylene oxide) ii. HDI	Ex vivo in rodent model for hydrogel injectability and gelation	** N/A	Controlled and triggered release drug	[64]
	Redox	A + C	PU Nanoparticles	i. PCL ii. HDI	i. MTS assay ii. NP (1000 µg/mL): Lung alveolar Type 1 cells (AT1) cell viability >80%	Degrade and reach 50% weight loss (polymers with increase of disulfide bonds) in 10 mM glutathione (GSH) after 14 days	Chemotherapy drug delivery	[56]
	Redox	A + C	PU micelles with mPEG block and PLA block with disulfide bonds mPEG-PUSS-mPEG	i. Copolymer PLA ii. Poly (ethylene glycol methyl ester) (MPEG) iii. Isophorone diisocyanate (IDPI)	i. CCK8 assay ii. Blank micelles (100 mg/L): HepG2 cell viability reduced to 76.3%; HUVEC cell dropped to 65%	Decompose within 24 h in the presence of 10 mM dithiothreitol (DTT)	Anticancer drug delivery	[4]
	Redox	A + C	PU with disulfide bonds, pendant carboxyl groups, and primary amine group PU-SS-COOH-NH_2_ micelles	i. PCL ii. PEG iii. HDI	i. CCK8 assay ii. Empty PU micelles (1 mg/mL): >80% cell viability for HUVEC and HepG2 cells	** N/A	Drug delivery	[59]
	Light	B + C	Serinol-based PU nanoparticles	i. 2- amino-1,3-propanediol ii. 4,5-dimethoxy-2-nitrobenzyl (4-nitrophenyl) carbonate	** N/A	Nanoparticle count rate decrease ~>30% after 15 min of UV irradiation	Nanocarrier for controlled drug release	[65]
	Shape memory	N/A	Shape memory polyurethane (SMPU)	Commercial SMPU, MM3520	** N/A	** N/A	Endovascular embolization	[66]
Dual	Shape memory + water	N/A	Thermoplastic PU/hydroxyethyl cotton cellulose nanofibers (TPU/CNF-C/CNTs)	Commercial TPU, BT-70ARYU	** N/A	** N/A	Sensors, actuators	[67]
	Thermo + light	A + C	PUA Nanoparticles	i. PCL ii. PLA iii. IDPI	i. VitaBright-48 (VB-48) assay, CCK8 assay ii. NIH3T3 cell viability ~70%	Weight loss of approximately >~10% in 28 days	3D cell-laden bioprinting	[68]
	Thermo + shape memory	A + C	PCL-based PU (PCLAU/Fe_3_O_4_)	i. PCL ii. Polytetramethylene ether glycol (PTMEG) iii. HDI	i. CCK8 assay ii. L929 fibroblast cell proliferation rate >80%	Weight loss of 67% after 13 weeks	Vascular stents	[69]
	Thermo + enzyme	B + C	Poly(ester urethane) Nanoparticles	i. 1,12-dodecane diol ii. L-tyrosine	i. MTT assay ii. WT-MEFs, HeLa, MCF7 cell viability results show non-toxic and biocompatible nature of the polymer	** N/A	Chemotherapy drug delivery	[60]
Multi	Thermo + shape + water	A + C	PU/nanoporous cellulose gel (PU/NCG)	i. PEG ii. TDI	** N/A	** N/A	Biomaterials, sensors	[70]

* PU Type: (A) isocyanate-based PU, (B) non-isocyanate-based PU, (C) PU synthesized with petrochemical-based polyols and (D) synthesized from bio-based polyols. ** N/A: information not available.

## 3. pH-Responsive Polyurethane

Table 3 summarizes all pH-responsive PUs reported in the past 10 years (Web of Science database, June 2021). The pH-responsive PUs are categorized into different applications: (A) biomedical and drug delivery; (B) optical imaging; and (C) biomaterials such as biosensors and bioactuators. pH-responsive PUs are favorable in drug delivery systems, particularly for the delivery of chemotherapy drugs, in which the pH-responsive PU carrier helps to modulate and improve the drug delivery by on-site targeting. The reported chemotherapy drugs used in pH-responsive PU systems are doxorubicin, paclitaxel and 5-fluorouracil, in which the PUs swelled at acidic pH (mimicking the environment of tumor pH) and demonstrated excellent drug release. The detailed applications and the mechanism of pH-responsive PUs reported will be further discussed in the following section.

Referring to Table 3, most of the reported pH-responsive PUs were produced by diisocyanates and polyols derived from petrochemicals. There were no studies reporting on non-isocyanate-based pH-responsive PUs created utilizing renewable resources, such as vegetable oil. As mentioned in the previous section, bio-based PUs have the advantage of protecting the environment and avoiding health concerns.

The reported acid-labile linkages include hydrazone and oxazoline linkages; hydrolysis of these linkages was reported to occur at pH ranging from 4.0–6.0. The reported basic ionizable moieties in pH-responsive PUs include 2-(diethylamino) ethyl methacrylate, 2-(diisopropylamino) ethyl methacrylate (DPA), 2-hydroxy ethyl piperazine (HEP), diethanolamine (DEA), N-methyldiethanolamine (MDEA) and pyridine; these PUs with basic ionizable moieties responded to pH environments ranging from 4.0–6.8. The reported acidic ionizable moieties include 2,2-dimethylol propionic acid (DMPA), lactic acid, glycolic acid, mercaptoacetic acid, sodium alginate, lysine, arginine and glutamine; most of these pendant groups respond to pH environments ranging from 7.4–10.4.

The reported dual-stimulus-responsive PUs with pH responsiveness are pH- and thermo-responsive, pH- and photo-responsive, pH- and shape memory-responsive, and pH- and redox-responsive. Within these dual-responsive PUs, pH- and thermo-responsive PUs and pH- and redox-responsive PUs have been reported for application in drug delivery systems. Multi-responsive PUs are introduced due to restriction of stimuli and applications in single-responsive PUs. The advancement of introducing multi-responsiveness is to exploit the maximum number of biological stimuli that occur at the tissue and intracellular level. For instance, the temperature and pH of the cancer tissue environment (40–42 °C and pH ≤ 6.8) are different from normal tissues (37 °C and pH = 7.4) [60]. Thermo-responsive polymeric materials undergo phase separation in the cancer tissue environment; these polymers selectively accumulate in this environment, leading to an increase of concentration. By introducing thermoresponsiveness along with pH responsiveness, finer modulation and specific target drug delivery could be achieved, since the parameters have increased.

**Table 3 polymers-14-01672-t003:** Different types of pH-responsive PUs with their attached functional response groups and their different applications.

Type of pH-Responsive PU	PU System	Reactants Used in Synthesis of PU	Ionizable Group/Linkage	Additional Stimulus Response	pH Responsiveness	* Applications	Reference	Reference Materials	Improvement to the Reference Materials
**Single**	PU micelles	i. PCL-Hydrazone-PEG-Hydrazone-PCL ii. LDI	Hydrazone linkage	N/A	pH ranging from 4.0–6.0, cleavage of hydrazone bond and degraded	A	[61,63]	N/A	N/A
	PU micelles	i. PCL ii. PEG iii. N-N-Dimethylacetamide (DMAc)	Hydrazone linkage	N/A	At pH 4.4, particle size increase due to swelling, drug release to 98%	A	[71]	Same PU without hydrazone bond	N/A
	PU/DEA copolymer	i. IPDI ii. Poly(propylene glycol) diacrylate (PPGDA)	2-(diethylamino) ethyl methacrylate	N/A	At pH 4.0, dynamic swelling and drug release	A	[11,72]	Same PU without DEA monomers	N/A
	PU micelles	i. PEG ii. PCL iii. IPDI	Diethanolamine	N/A	At pH 5.5, highest drug release	A	[73]	N/A	N/A
	PU copolymer	i. PEG ii. 1,6-haxanediol (HD) iii. HDI iv. MDI	HEP	N/A	At pH 4.5, swelled twofold and close to zero drug release; however, sodium diclofenac incorporated release at elevation to pH 7.0	A	[3,74]	PEG-HD-MDI-HD without HEP monomers	Reversible and sharp switch between “on” and “off” drug release, serves as window membrane in reservoir-type intravaginal rings
	PU copolymer NPs	i. MPEG ii. IPDI	i. HEP ii. DMPA	N/A	PU containing higher HEP ratio, swelled and highest drug release at pH 5.0	A	[75]	N/A	N/A
	PU copolymer hydrogel	i. PEG ii. HDI iii. PG	DMPA	N/A	Drug release at pH 7.0	A	[76]	N/A	N/A
	PU hydrogels-	i. PVA ii. Chitosan iii. TDI	Poly(azomethine-urethane) (PAMU)	N/A	Highest swelling degree at pH 3.0; increase of PAMU, swelling degree increase, release of drug increase	A	[77]	N/A	N/A
	PU nanomicelles	i. PEG ii. IPDI iii. Poly(neopentyl glycol adipate) diol (PNA-2000) iv. HEMA	2-[N,N-bis (2-hydroxy-ethyl)] aminoethanesulfonic acid sodium salt (BES-Na)	N/A	Drug release rate: pH 5.0 > pH 6.8 > pH 7.4	A	[12]	N/A	N/A
	PU-sodium alginate (SA) blend	i. Bis-hydroxyethylene terephthalate (BHET) ii. PEG iii. HDI	Sodium Alginate	N/A	Swelled at pH 7.4, sustained and prolonged release of incorporated protein or insulin	A	[78,79]	N/A	BHET derived from PET waste, biocompatible
	Cellulose crosslinked PU	i. PCL ii. HDI	i. Lactic acid (LA) ii. Glycolic acid (GA) iii. DMPA	N/A	All 3 PUs swelled and highest release of incorporated drugs at pH 7.4	A	[80]	N/A	Control release rate by changing chain extender; Drug release rate LAPU > DAPU > GAPU
	PEG-HTPB (g-COOH)-PEG triblock copolymer	i. Hydroxyl-terminated polybutadiene (HTPB) ii. MPEG iii. HDI	Mercaptoacetic acid	N/A	At pH 7.4, micelles swelled rapidly and released drug	A	[81]	N/A	N/A
	PU/cellulose acetate phthalate (CAP) fibers	Commercial PU	CAP	N/A	Rapid release of Rhodamine B at pH 7.4 within 1 min	A	[82]	- Pure CAP - Pure PU fibers	Improved tensile strength compared to previously reported CAP fibers
	PU films	i. PEG ii. HDI	i. Lysine ii. Arginine iii. Glutamine	N/A	PU-Arginine shows highest drug release at pH 4.4; All PU shows average drug release of 64% at pH 10.4	A	[83]	N/A	N/A
	PEG-PU copolymers	i. PEG ii. HDI	i. HEP ii. DMPA	N/A	Highest pH buffering capacity 7.02, specific responsiveness not mentioned	B	[84]	N/A	N/A
**Dual**	PU	i. PEG ii. MDI	i. DMPA ii. MDEA	Thermo-responsive	PU-MDEA: Swells at pH 4.0–5.5 PU-DMPA: Swells at pH 8.5–10.0	N/A	[85]	N/A	N/A
	PU Micelles	i. PCL ii. MDI	i. HEP ii. MDEA iii. N-butyl diethanolamine (BDEA)	Thermo-responsive	HDI-MDEA and HDI-BDEA, rapid drug release at pH 4.0	A	[86]	N/A	N/A
	PU/DPA	i. PPGDA ii. PEGMA iii. IPDI	DPA	Thermo-responsive	Increase of DPA, results in highest swelling degree at pH 4.0	A	[87]	Same PU without addition of DPA/PPGDA/PEGMA mixture	N/A
	PEG-PCL based PU blend with cellulose nanocrystals (CNC)	i. PEG ii. PCL iii. IPDI	i. Pyridine-4-carbonyl chloride (-C_6_H_4_NO_2_) ii. 2,2,6,6-tetramethyl-1-piperidinyloxy (-COOH)	Shape memory	CNC-COOH; At pH 4.0, folded strip of CNC-C_6_H_4_NO_2_ recovers to straight	C	[88]	N/A	N/A
	PU	i. PPG ii. IPDI	Pyridine	Shape memory	Swells at pH 1.3, drug release and shape recovers	A, C	[10]	N/A	N/A
	Azo-cationic waterborne polyurethane (CWPU)	i. PEG ii. PCL iii. MDI	i. MDEA ii. Azobenzene	Photo-responsive	Shows different color in different pH medium	B	[89]	N/A	N/A
	PU micelles with disulfide linkage	i. PEG ii. PCL iii. LDI	MDEA	Reduction-responsive	Rapid drug release at pH 5.5	A	[90]	N/A	N/A
	PU with disulfide bonds	i. PCL ii. HDI iii. Bis (2-isocyanatoethyl) disulfide (CDI)	Poly(2-ethyl-2-oxazoline)(PEOz)	Reduction-responsive	Drug release rate higher at pH 5.0	A	[91]	- End-group-carboxylated PEOz-PLA - PEOz-hydrazone-DOX	Cumulative drug release increase with presence of 1, 4-dithio-D, L-threitol (DTT)
	MPEG/PU triblock copolymers with disulfide linkage	i. MDI ii. HDI iii. MDEA iv. BDEA	Bis-1,4-(hydroxy-ethyl) piperazine (HEP)	Reduction-responsive	pH 5.5 and 6.8, swells and faster drug release	A	[92,93]	N/A	N/A
**Multi**	PU	i. PEG ii. MDI	DMPA	Thermo-responsive and shape memory	For PEG-30%-MDI-DMPA, fixes deformed shape at pH 2.0, recovers shape at pH 9.0	N/A	[94]	N/A	N/A

* Applications: (A) Biomedical and drug delivery, (B) optical imaging, and (C) biomaterials.

### 3.1. Applications of pH-Responsive Polyurethanes in Drug Delivery Systems

The polymeric material used in a drug delivery system (DDS) must be capable of delivering the active ingredients to the desired site of action in response to on-site pH stimulus, and the materials must be composed of non-toxic, biocompatible and biodegradable compounds. pH-responsive polymers can provide targeted delivery or controlled delivery of drugs to specific parts of the body with a specific release rate, in order to achieve optimal therapeutic efficiency while at the same time reducing adverse effects and toxicity [1,72].

pH-responsive PUs have been investigated for their potential applications in drug delivery systems, owing to their ease of synthesis, ability to form stable nanocarriers with H-bonding to the backbone [95], excellent mechanical properties [82], biocompatibility and biodegradability. pH-responsive PUs have been explored in various routes of administration, namely oral, intravaginal and intravenous. Yu et al., (2012) reported on pH and reduction PU micelles loaded with doxorubicin (DOX), in which drug release was observed in an acidic environment (pH 5.5) and showed low toxicity in vivo [93].

#### 3.1.1. Oral Administration

Oral administration is the most frequently used administration route for most solid and liquid forms of drugs, due to safety and patient compliance. The main challenge of oral formulations is the low aqueous solubility of hydrophobic drugs. In addition, the drug compounds must be stable and maintained in their active form when they reach the absorption site. There are several factors influencing the absorption of drugs, which include the harsh pH condition of gastric fluid, variation in pH condition and enzymatic reaction throughout the gastrointestinal (GI) tract, food and hepatic metabolism [81,96]. To achieve drug absorption at intestinal sites, drugs could be coated with polymer film or conjugated into the polymer backbone to bypass the potential degradation caused by gastric fluid. Due to different biological barriers, enzymes and pH environment throughout the GI tract, designing a suitable polymeric-coated drug delivery system is still challenging to optimize the formulation at its desired release rate and site. Moreover, biodegradability and biocompatibility are the main concerns for drug delivery systems, as there may be unpredictable interactions of synthetic materials or degradation byproducts with human body cells [97,98]. The delivery material, particle size and polydispersity index of the polymers ought to be taken into consideration, as they might influence delivery, efficacy and toxicity. The monitoring of release dose, release rate, on-site specific drug release, drug loading capacity and cytotoxicity should be studied precisely [99].

Nabid et al., (2016) reported on a novel one-pot synthesis of a pH-responsive hydroxyl-terminated polybutadiene (HTPB)-based PU nanocarrier for oral delivery of a hydrophobic drug, ibuprofen (IBU), with carboxylic acid groups (–COOH) being introduced into the PU structure to gain pH responsiveness [81]. IBU was loaded into the polymer nanoparticles via nano-precipitation method. The PEG acts as the hydrophilic blocks, while the HTPB acts as the hydrophobic blocks, which leads to self-assembly into micelles in the water (pH 5.0). The drugs were entrapped in the core of the micelles. Increasing the carboxylic acid content in the polymer backbone enhanced the loading capacity and efficiency due to better stability of the micelles formed. The micelles swell at pH 7.4 and further dissociate at pH 9.0 (Figure 10). This is due to the increased repulsion charge in the deprotonate carboxyl group, which reduces the hydrophobic chain attraction and causes swelling. The in vitro drug release was conducted at pH 2.0, pH 6.8 and pH 7.4. At pH 7.4, the micelles swelled rapidly, causing a high diffusion rate of IBU into the environment.

Furthermore, by adding carboxylic acid groups into the polymer, it is potentially suitable for oral drug delivery systems, as it could aid in protecting the drugs from acid degradation and successfully deliver it to the intestine for absorption and distribution. The –COOH groups in pH < 6 (gastric fluid environment) are protonated and hence decreased the solubility of the micelles, while the –COOH group ionized in pH > 6 (intestinal fluid environment), increased the solubility of the micelles [96].

In another work reported by Bhattacharyya et al., (2014 and 2016), pH-sensitive polyurethane-sodium alginate (PU-SA) was blended with poly (ethylene terephthalate) waste and PEG used as chain extender [78,79]. The swelling behavior of PU-SA crosslinked with calcium chloride was studied under different pH environments (pH 1.2 mimics gastric fluid; pH 7.4 mimics intestinal fluid), in which the polymer beads showed maximum degree of swelling (more than 800% for all ratios of PU-SA beads) at pH 7.4. The hydroxyl groups of PU and alginate were deprotonated, causing electrostatic repulsion and cross-linked disruption, hence increasing the degree of swelling. The PU-SA was then loaded with bovine serum albumin (BSA). As a result, the BSA-conjugated PU-SA beads with the highest PEG content showed highest release of BSA at pH 7.4. The incorporated BSA was released through a continuous erosion mechanism, in which the dissolution fluid penetrates into the beads, leading to a swollen matrix; the matrix further disintegrates to release the drugs. The formulation of PU-SA beads with the highest PEG and SA concentration possessed better encapsulation efficiency due to the increased binding sites for Ca^2+^ ions, which results in a denser crosslinked gel structure that could increase encapsulation efficiency. That PEG could enhance drug-loading capacity was also reported in a study conducted by Chandel et al., (2016) [100]. The PU could be applied as a polymeric carrier to entrap oral-administered drugs or proteins which are targeted to be released at the small intestine. However, in vitro cytotoxicity tests and in vivo evaluation should be performed to further confirm its biocompatibility.

Polo Fonseca et al., (2018) reported on the synthesis of amphiphilic crosslinked PU hydrogels produced by PEG and PCL-triol as polyols and incorporated with pH selectivity drugs [2]. Interestingly, these amphiphilic PU matrices were not functionalized by or grafted with any acidic or basic pendant group. Sodium diclofenac, a non-steroidal anti-inflammatory drug, was incorporated into the PU for a drug release study. In this study, cumulative drug release was performed to mimic the administration route. At pH 1.6, the sodium diclofenac-loaded PU was swollen, while the drug precipitated and only released into the medium when the pH was increased to 7.4. When the PU hydrogel was swollen, the solution inside the hydrogel reached 4.15 (the pKa of sodium diclofenac), hence causing the drug to diffuse into external media. In short, this synthesized PU presented high-efficiency loading and release of acidic hydrophobic drugs at neutral or basic conditions, with sustained rate up to 40 h, indicating its potential application in delivery of nonsteroidal anti-inflammatory drugs.

Summarizing the reported works, drug release studies were basically performed and observed to happen at pH 7.4 (mimicking the intestinal pH environment); while at pH 1.2 (mimicking the gastric pH environment), near to zero drug release was observed, which implies that the PU protected the loaded drugs from gastric fluid attack. Most of the reported PUs as carriers for drugs meant for oral dosage form are high pH-responsive polymers, which fulfil the requirement to release drug content in the intestinal area. However, there are limited reports on in vivo evaluation of the pH-responsive PUs used as carriers for oral dosage forms. After ex vivo or in vivo studies, the distribution and diffusion of polymers and the incorporated drugs can be investigated throughout different parts of the body.

#### 3.1.2. Intravaginal Administration

Intravaginal administration is used to achieve local therapeutic administration; the absorption of drugs occurs through the vaginal epithelium, bypassing the hepatic metabolism. The normal vaginal pH of healthy women is at the range of 3.8 to 4.2 due to lactic acid production by natural vaginal microflora [82]. When semen is present after sexual intercourse, the vaginal pH will be increased to slightly alkaline conditions ranging from pH 7.0 to 8.0. Thus, pH-responsive contraceptive formulations have been developed.

Hua et al., (2016) reported on the development of a PU/cellulose acetate phthalate (CAP) coaxial fiber as potential carrier for intravaginal delivery of contraceptive drugs [82]. The PU served as the core, while the CAP acted as the shell of the fibers. The CAP was reported to be minimally soluble in healthy vaginal flora, but highly soluble when exposed to a pH of approximately 7.0. Rhodamine B, a fluorophore dye, was added to the PU for a pH-dependent release study. Rhodamine B remained loaded in the fibers at pH 4.3 (simulated vaginal fluid) while it was rapidly released at pH 7.4 (simulated vaginal fluid with presence of semen).

Solanki et al., (2015) reported on the synthesis of cellulose-crosslinked waterborne PU by reaction of PCL diol with hexamethylene diisocyanate (HMDI). pH-responsive groups including lactic acid, glycolic acid and DMPA were introduced into the PU films [80]. A model drug, felodipine, was loaded in all PUs, in which the drug release rate with respect to pH followed the order 7.4 > 4.5 > 1.2. The acid chain extenders (anionic groups) were added so that it was ionized im basic conditions, increasing the charge density of the polymer and expanding the network for easier penetration of the external solution, resulting in a higher degree of swelling. The pH-sensitive PUs were proposed for vaginal- or colon-specific drug delivery systems.

Intravaginal ring (IVR) technology has often been applied for continuous dosing of contraceptives and hormone therapies. It has advantages over other formulations as they can provide sustained control and prolonged release of drugs. Administration of anti-HIV drugs and drugs for treatment of sexually transmitted diseases (STDs) through this route is being considered. The materials used for vaginal rings should be flexible, inert, non-irritating, biocompatible and not interfere with the balance of vaginal microflora [76,101].

Kim et al., (2018, 2017) reported on the production of PU copolymers for the development of a pH-responsive membrane attached to intravaginal rings for the prevention of HIV transmission [3,74]. The PUs were incorporated with the pH-sensitive moiety, 1,4-bis(2-hydroxyethyl) piperazine (HEP). The results from SEM and swelling degree examination showed that the PU copolymer swelled at the highest degree at pH 4.5, and remained at equilibrium after pH 7.0. This is due to the protonation of tertiary amine groups of HEP at pH 4.5, causing high ionic repulsion and expansion between the neighboring polymer chains (Figure 3). Sodium diclofenac was used as the model drug to mimic the properties of anionic anti-HIV drugs (betulinic acid and bevirimat). Figure 11 illustrates the utilization of the PU membrane as a “window” membrane in the reservoir of IVR. The PU membrane has achieved a reversible and sharp switch between “on-and-off” drug release, serving as on-demand delivery to prevent STDs. Close to zero drug release was observed at pH 4.5, due to the swollen fibers blocking the pathway for the drug molecules to be released from the IVR reservoir. Drug release is only observed when the pH elevates to 7.0 (mimicking the environment of semen after sexual intercourse) (Figure 11).

In another study from the same authors, dimethylolpropionic acid (DMPA) was used as a pH-sensitive molecule [76]. The pH-sensitive PU loaded with DMPA, i.e., PEG-DMPA-HDI-PG, was synthesized to encapsulate nanoparticles as a physically crosslinked hydrogel within a segmented-reservoir IVR. PEGylated poly(aspartic acid)-based copolymer conjugated with the fluorescent dye orange II (PASP-PEG-Ph-Orange) was synthesized to self-assemble in aqueous solution to form nanoparticles for the release study. The nanoparticles with fluorescent orange dye were developed for easy tracking of the nanoparticles in the release study.

High release of PASP-PEG-Ph-Orange was observed at pH 7, due to weaker hydrophobic interaction between the copolymer chains and space between the particles (Figure 12). The carboxylic groups of DMPA ionized, causing the polymer to swell and causing lower hydrophobic interaction, which expanded the space between the copolymer chains, thus inducing the release of the entrapped nanoparticles (Figure 4).

The reported pH-responsive PUs have shown their potential applications in intravaginal delivery systems. For intravaginal administration of contraceptive drugs, high pH-responsive PUs are more fitting for the release of contraceptive drugs in a higher pH environment after exposure to semen. For all of these studies, clinical studies are required to further investigate the efficacy and safety aspects of the intravaginal delivery system.

#### 3.1.3. Intravenous Administration for Chemotherapeutic Drugs

Drugs given through intravenous (IV) administration enter the systemic circulation directly to treat emergent concerns. Drugs or substances which are administered through the IV route include proteins, antibiotics, pain medications and chemotherapy drugs. Due to the non-selectivity of chemotherapeutic drugs, it is desirable to incorporate them in polymer carriers to target cancer cells, thereby reducing adverse effects and improving the therapeutic efficiency of anticancer drugs.

Chemotherapy is the most-often used strategy for cancer treatment. However, there are a few factors to be considered, including non-specificity, wide biodistribution, low concentration in tumor tissue and systemic toxicity [93,102]. As a result, normal cells could be exposed to the cytotoxic effects of the chemotherapeutic drugs. The side effects caused by chemotherapeutic drugs are for instance nausea, immune suppression, anemia, hepatoxicity, nephrotoxicity and death. To overcome these problems, on-site targeted delivery and sustained delivery should be achieved. Hence, pH-responsive polymeric drug carriers were explored to enhance the release of anti-cancer drugs by responding to the specific acidic microenvironment of tumors (average pH 6.0) [95].

Doxorubicin, a first-line hydrophobic anticancer drug, has been widely used in the evaluation of drug-release studies for chemotherapy. Long et al., (2016) reported on the synthesis of coumarin-based cross-linked PU micelles with a pH-responsive hydrazone bond. Doxorubicin (DOX)-loaded PU showed high release at pH 5.8 due to the hydrolysis and cleavage of hydrazine bonds in an acidic medium (Figure 13) [103]. Liao et al., (2018) reported on synthesis and preparation of DOX-loaded PEG-grafted PU micelles [104]. At pH 6.0, the micelles demonstrate high release due to the specific intermolecular interactions between the DOX molecules and aromatic groups of the PU backbone, which rapidly dissociated under low pH. The PEG-grafted PU demonstrated enhanced properties, including low critical micelle concentration (CMC), low cytotoxicity, tunable drug loading capacity and high encapsulation stability. He et al., (2016) reported on the synthesis of PU copolymers containing different ratios of poly(ethylene glycol) methyl ether (MPEG), carboxylic acid groups and HEP groups [75]. The DOX-loaded pH-responsive PU micelles with the highest HEP content and lowest carboxylic acid content showed the highest drug release in a slightly acidic environment (pH 5.0 and 6.5).

Shoaib et al., (2018) reported on the synthesis of PU films by polyethylene glycol with HDI in the presence of different amino acids as chain extenders [83]. The amino acids, such as lysine, arginine, and glutamine were used to offer pH-responsive properties to the PU. An anti-cancer drug, imatinib, was loaded into the PU films for drug release study at pH 4.4, 7.4 and 10.4. As a result, the highest drug release was observed at pH 4.4 due to the protonation of amino groups in the arginine chain extender causing the electrostatic repulsion between chains, hence the swelling and solvent penetration increased. However, the pH value investigated in this study was broad, which might not be benchmarkable with other studies. Further investigation of pH values ranging from pH 5 to pH 7.4 is recommended to explore diverse applications of the PU films.

In short, it is observed that low pH-responsive PUs or PUs with acid-labile linkage polymers could be candidates for chemotherapy drug delivery. This assumption is also supported by the summary made in Table 1, where generally, most of the reported pH-responsive polymers are incorporated with acid-labile linkages. With the mechanism of cleaving the acid-labile linkages in the low pH environment of the tumor cells, anticancer drugs can be delivered and targeted to the tumor cells, improving bioefficacy and reducing the toxicity of the drugs.

#### 3.1.4. Controlled Drug Delivery

Controlled drug delivery is a predesigned system which delivers the drug at a predetermined rate, locally or systemically, for a specified period of time [106]. Controlled drug delivery systems facilitate the protection of fragile drugs, maintenance of drug levels within desired range, fewer administrations to increase patient compliance, biocompatibility and improved bioavailability at the site of action by avoiding first-pass metabolism [107]. Polymers were being used in drug delivery systems to enhance drug stability and modify drug release properties. The release of drugs may be constant over an extended period or may be triggered by the external environment of the body.

The development of controlled drug delivery systems includes nanocarriers such as micelles, vesicles, solid lipid nanoparticles, and liposomes to aid the delivery of drugs. Nanocarriers help in improving dispersion of hydrophobic drugs in the human body and enhancing delivery efficiency and uptake. In addition, they also protect drugs from acidic conditions which might degrade their bioactivity. The commonly used pH-responsive polymers for controlled drug delivery systems are the crosslinked polymer networks, such as microgels and hydrogels. Hydrogels possess porous structures and good ability to swell with water. Most of the pH-responsive polymers used in controlled drug release are dual pH- and redox-responsive polymers, containing either disulfide or ketal crosslinkers which will undergo cleavage of bonds in different pH gradients. These crosslinkers protect the loaded drugs, and control release at specific acid pH and oxidation responsiveness.

Cheng et al., (2016) reported on simple preparation of diselenide-crosslinked PEGylated PU nanogels with pH- and oxidation-responsive properties, containing MDEA as the pH-responsive functional group [108]. The nanogels were reported to have superior colloidal stability due to the diselenide crosslinking. The loading efficiency of drugs with nanogels was reported to be 76.3%, which was higher than that of non-crosslinked micelles. In addition, the compact inner core of nanogels helped to prevent possible leakage of drugs during the dialysis process. Accelerated drug release was observed in the presence of high H_2_O_2_ concentration and pH 5.0, due to its swelling behavior under acidic conditions as well as the protonation of tertiary amines, promoting rapid diffusion into the nanogels (Figure 14).

The same research group also reported on one-pot condensation polymerization of multi-block copolymer poly (ether urethane) (PEU) incorporated with α-cyclodextrin (α-CD) containing diselenide bonds and DMPA [109]. This amphiphilic PEU copolymer was reported to self-assemble into nanoparticles in aqueous solution without addition of organic solvent, which was found suitable for loading hydrophobic drugs. The hydrogel was reported to have shear thinning properties, which could be used as an injectable hydrogel matrix to carry drugs. pH-induced reversible sol-gel phase transition was also observed between pH 6.8 and 9.2 in the PEU system. In a high pH medium, the carboxyl moieties of DMPA were deprotonated and enhanced hydrophilicity, hence the nanoparticles were swollen and turned to a gel-like composition. At the same time, by increasing the concentration of H_2_O_2_, an oxidation reaction occurred, and the polymer degraded to trigger drug release (Figure 15). In contrast to the previous study, the PEU system is a high pH-responsive polymer.

Song et al., (2018) developed a PU-based nanomicelle with a hydrophobic soft chain of poly(neopentyl glycol adipate) diol (PNA-2000), hydrophilic soft chains of PEG, chain extender 1,4-butanediol (BDO) and 2-[N,N-bis (2-hydroxyethyl)] aminoethanesulfonic acid sodium salt (BES-Na) as the amphoteric functional moiety [12]. Folic acid, a hydrophobic model drug, was incorporated into the micelles through the dialysis method. The PU micelles were aggregates and self-assembled at very low concentration, which improved the micellar stability. In an acidic environment (pH 5.0 and 6.8), drug release was observed to be higher than that in a neutral environment (pH 7.4), due to the protonation of tertiary amine groups in BES-Na. Pre-clinical studies were planned by the authors.

The reported studies have shown PU as a promising candidate for controlled drug delivery applications, owing to PU having highly tunable properties, for which it is suitable in incorporating different linkages and functional moieties into the backbone. For instance, by conjugating environment-labile linkages such as diselenide groups, the PU could achieve faster degradation through oxidation, compared with traditional PU [109]. In addition, the reported PU shows responsiveness in specific ranges of pH environments and drug release rate. The results have shown triggered release at specific pH, prolonged release of drug until 17 h and more. In both studies from Cheng et al., the drug release data of the nanoparticles were fitted into the Ritger–Peppas equation, which shows different drug release mechanisms in different conditions during certain hours. With the dual responsiveness towards pH and oxidation conditions, fine-tuning of various parameters could be used to achieve desired release profiles. However, in vitro biodegradation tests and cytotoxicity assays were not reported in all the studies discussed. Hence, biocompatibility and hemocompatibility have yet to be confirmed. Furthermore, in vivo drug release studies could be further conducted to monitor the bioavailability of the polymer-incorporated drugs at specific sites of action.

### 3.2. Applications of pH-Responsive Polyurethanes as Biomaterials

Biomaterials are non-viable substances, naturally or synthetically derived, used in biological systems and parts of medical devices. These materials must possess optimal characteristics such as good stability, biocompatiblity, biodegradability and desired mechanical properties. Stimuli-responsive polymers have been widely studied in a broad range of smart biomaterials, such as actuators and sensors.

Wu et al., (2014) have prepared chitosan-modified cellulose whiskers/thermoplastic PU (CS-CW/TPU) composites with shape memory-responsive and pH-responsive properties [110]. By comparing chitosan-modified CW/TPU with pristine CW/TPU, it can be concluded that chitosan helps in enhancing the degree of water absorption, altering the pH dependence of CW/TPU composites and their mechanical adaptive properties. CWs are rich in sulfate-ester groups, which give the properties of low wettability and low mobility of chain segments in an acidic medium. By modifying the surface with chitosan, the modulus contrast was improved in acidic environments while maintaining the high modulus contrast with basic solutions. Chitosan was being protonated in acidic mediums, which enhanced the ability of the TPU to swell. In the acidic environment, the CS-CW/TPU showed better mechanically adaptive properties, undergoing larger shrinkage and greater contractile forces (up to 0.11 N); while in the basic environment, due to the negatively charged hydroxide groups and residual NaOH between CWs, it impeded the shrinkage and lowered the contractile forces. The force-generation ability indicated that the CS-CW/TPU produced has potential application in water-sensitive switches, water-active actuators, sensors, and ultralow-power generation.

Similarly, Li et al., (2015) developed a novel pH-responsive shape memory poly(ethylene glycol)-poly(ε-caprolactone)-based polyurethane (PECU) nanocomposite with modified cellulose nanocrystal (CNC) percolation networks [88]. The CNC, functionalized with carboxyl groups (CNC-COOH), underwent deprotonation at high pH conditions, leading to electrostatic repulsion, and formed a monodisperse emulsion; whereas pyridine moieties (CNC-C_6_H_4_NO_2_) protonated at low pH condition (pH < 5.0) leading to disassociation of hydrogen bond interaction and increased hydrophilicity. The shape memory properties were dependent on the pH responsiveness of PECU, which can act as a switch unit by controlling the hydrogen bonding interaction with alteration of pH. The PECU with CNC-C_6_H_4_NO_2_ moiety was reported to be able to recover its original shape after immersion in a low-pH environment for 30 min (Figure 16). The PECU was proposed to be used in biomaterials, smart sensors or actuators.

However, in the above two studies, no in vitro studies were conducted to confirm their biocompatibility and safety for use.

### 3.3. Applications of pH-Responsive Polyurethanes in Optical Imaging

At present, the common imaging methods used in biomedical applications include positron emission tomography (PET), computed tomography (CT), magnetic resonance imaging (MRI) [111,112] and fluorescent imaging [112,113]. Optical imaging has been increasingly applied in medical fields to obtain detailed images of inner body segments. Different from other radioimaging, optical imaging uses visible light such as ultraviolet and infrared light, and the special properties of photons (fluorescence and luminescence) to capture images of cells and molecules. However, the current technologies for molecular imaging, such as PET, have limitations on their spatial resolution and incur radiation damage. By introducing fluorescent molecules into polymer-based carriers, they can act as imaging probes for monitoring and diagnosis; at the same time the polymeric carriers must be biocompatible, of low toxicity and have long-term stability [114,115,116].

The application of pH-responsive PU in diagnostic imaging properties is mainly as a self-fluorescent polymer. Xi et al., (2017) reported on a pH-responsive self-fluorescent methoxy polyethylene glycol (MPEG)-terminated PU multi-block copolymer via polycondensation reaction [117]. 1,4-Bis(hydroxyethyl)piperazine (HEP) containing amino groups was conjugated into the micelles to offer pH responsiveness. As the ratio of HEP increased, the pH buffering capacity of the PU increased, until it was close to the pH value of tumor tissues (pKa 6.65, 6.72, 6.8 and 7.02), which helped to reduce the burst release of the drug and introduced sustained release at the tumor site.

Fluorescent dye fluorescein isothiocyanate (FITC) was conjugated into the PU to confirm its pH-dependent fluorescence property. When the pH was increased to neutral or alkaline (pH 7 and pH 10), the lactone structure of FITC produced strong UV-Vis absorbance peaks. In addition, the carboxyl groups ionized and caused the molecular chain to extend, resulting in increased fluorescence intensity. At lower pH (pH 3 and pH 5), the lactone ring of FITC underwent a ring-opening process, which decreased the fluorescence intensity. The author concluded that the pH-responsive self-fluorescent PU demonstrated on-and-off fluorescence behaviors responding to minor alterations of pH, hence it possessed a potential application in biological imaging.

Similarly, Xia et al., (2016) reported on the production of pH-responsive PEG-PU multi block copolymers via polyaddition reaction [118]. HEP was conjugated as the pH-responsive segment, while iron oxide nanoparticles (Fe_3_O_4_) and fluorescent dye FITC were incorporated into the PU to evaluate its potential application in MRI and optical imaging. The trend of results for pH-dependent fluorescence study was the same in both simulated studies. At pH 7.4, a high loading efficiency of 75% Fe_3_O_4_ (contrast agent used in MRI) was achieved. The PEG-PU conjugated with HEP exhibited its potential as a dual-modality optical imaging probe.

The PU produced in the above studies has passed the in vitro cytotoxicity test, which makes it one step closer to being a smart vehicle in optical imaging. In vivo biocompatibility studies should be conducted to further prove the potential applications of PU in optical imaging.

## 4. Future Perspectives and Conclusions

The studies highlighted in this review show great results pointing to pH-responsive PUs enhancing drug delivery systems for intravaginal administration, oral administration, and intravenous administration of chemotherapeutic drugs and controlled drug delivery. The pH responsiveness of the versatile PUs attracts attention in the pharmaceutical and biomedical industries. Although having great potential in biomedical and drug delivery systems, there is still much effort required to confirm their effectiveness and safety in biological systems. Generally, in most of the reported works on pH-responsive PU systems for biomedical and drug delivery applications, the stability, biocompatibility, biodegradability and mechanical properties of the PU as the potential polymeric base for various potential drug delivery and biomaterials were not evaluated, discussed or compared with other polymeric materials. In addition, most of the reported pH-responsive PUs were produced from non-renewable petrochemical-based polyhydroxyl compounds and the toxic polyisocyanates. Avoidance of using toxic reactants and solvents in the production of PUs should be promoted to ensure the safety and biodegradability of PUs to safeguard user safety and environmental preservation. In vitro and in vivo studies of pH-responsive PUs should be conducted accordingly to pave the way for further clinical usage.

## Data Availability

Not applicable.

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
