# Peer review of "PH Responsive Polyurethane for the Advancement of Biomedical and Drug Delivery"

_polymers, 2022, doi:10.3390/polym14091672_

Round 1

Reviewer 1 Report

Dear author, please revise your manuscript to the following suggested points

I strongly recommend revising this manuscript as follows:

  1. I would like to suggest to the authors, the lipase enzyme responsive ester linkages are very important so that some examples should be added to the manuscript in lines number 263,264, author should cite these articles in the manuscript Effect of Polyethylene Glycol on Properties and Drug Encapsulation–Release Performance of Biodegradable/Cytocompatible Agarose–Polyethylene Glycol–Polycaprolactone Amphiphilic Co-Network Gels https://doi.org/10.1021/acsami.5b10675, and Degradable/cytocompatible and pH-responsive amphiphilic conetwork gels based on agarose-graft copolymers and polycaprolactone https://doi.org/10.1039/C5TB01251A
  2. In the segment of pH-responsive polymers, the author should discuss very new chemistry recently reported nucleophilic substitution reaction, that showed very good pH and temperature responsiveness, author should cite those articles in the manuscript. Reactive compatibilizer mediated precise synthesis and application of stimuli-responsive polysaccharides-polycaprolactone amphiphilic co-network gels https://doi.org/10.1016/j.polymer.2016.07.033 Dually crosslinked injectable hydrogels of poly (ethylene glycol) and poly [(2-dimethylamino) ethyl methacrylate]-b-poly (N-isopropyl acrylamide) as a wound healing promoter DOI https://doi.org/10.1039/C7TB00848A Self-Assembly of Partially Alkylated Dextran-graft-poly[(2-dimethylamino)ethyl methacrylate] Copolymer Facilitating Hydrophobic/Hydrophilic Drug Delivery and Improving Conetwork Hydrogel Properties https://doi.org/10.1021/acs.biomac.8b00015
  3. The author should discuss the commercially available drug delivery system/device based on pH responsiveness behaviour.

  1. I would like to recommend to the authors should cite Nanosystems for drug delivery of coenzyme Q10 in the drug delivery part of the manuscript.

  1. The author should add information in the manuscript about other applications of ph responsive materials, for instance, pH-responsive materials are very important in the field of biomedical applications they are used in drug delivery as well as several other applications such as postoperative antiadhesion applications. https://doi.org/10.1002/mabi.202000395

Author Response

Thank you for all the comments, we have gone through the comments thoroughly and made necessary amendments on the manuscript. Please find the revision report as enclosed.

Reviewer 2 Report

The manuscript mainly focused on the applications of pH responsive polyurethane in the biomedical and drug delivery fields for the recent years and shortly discussed the applications as biomaterials and in optical imaging. And the manuscript reviewed the developments of pH responsive polyurethane on oral administration, intravaginal administration, intravenous administration for chemotherapeutic drugs and controlled drug delivery.

Works reviewed in this manuscript showed good results of pH responsive polyurethane when applied to the drug delivery system for intravaginal administration, oral administration, intravenous for chemotherapeutic drugs and controlled drug delivery. Meanwhile, pH responsive polyurethane had great potential in biomedical and drug delivery fields. This manuscript could be accepted after necessary corrections and the specific suggestion was listed as follows.

  1. The summary of polyurethane synthesis in section 2.1 in the page 8 to 9 could be linked to the pH responsive polyurethane in section 3. For example, did the different synthesis methods of polyurethane affect the pH responsiveness of polyurethane? If so, what would be the impact? And if mentioned in the references, what was the synthesis methods of the polyurethane mentioned in section 3?
  2. The manuscript mentioned that polyurethane had good biocompatibility several times. What about the biocompatibility after the release of incorporated substances in the drug delivery system?
  3. In the page 12, line 292 to 294, the manuscript mentioned four different applications of pH responsive polyurethane, while section 3 described only three of these applications in detail. The application of bioengineering can also be explained briefly.
  4. Many of the figures in the manuscript were vague, such as Figure 12 in the page 20. It would be better to change to clearer figures.
  5. In the page 17, Figure 10 quoted from Ref. 77 was distorted due to a change in aspect ratio.
  6. It was a better choice to unify the format of the structural formulas given, for example, Figure 8 can be unified with others.
  7. If the figure was taken from other papers, the corresponding reference should be labeled below, for example, Figure 13 in the page 21 should be supplemented with its Ref. 98.
  8. The authors could consider adding the following review articles into references which would again increase the interest to general simuli-responsive smart biomaterial readers: Chemical Society Reviews‚ 2021, 50, 8319-8343; Advanced Functional Materials‚ 2022‚ 32, 2108749; Biomaterials Science‚ 2020‚8, 4940-4950.

Author Response

(The authors gave the same response as above.)
